# Endothelin System and Ischemia-Induced Ventricular Tachyarrhythmias

**DOI:** 10.3390/life12101627

**Published:** 2022-10-18

**Authors:** Eleni-Taxiarchia Mouchtouri, Thomas Konstantinou, Panagiotis Lekkas, Theofilos M. Kolettis

**Affiliations:** 1Department of Cardiology, Medical School, University of Ioannina, 451 10 Ioannina, Greece; 2Cardiovascular Research Institute, 451 10 Ioannina, Greece

**Keywords:** myocardial ischemia, ventricular tachyarrhythmias, sympathetic activation, endothelin

## Abstract

Despite the contemporary treatment of acute coronary syndromes, arrhythmic complications occurring prior to medical attendance remain significant, mandating in-depth understanding of the underlying mechanisms. Sympathetic activation has long been known to play a key role in the pathophysiology of ischemia-induced arrhythmias, but the regulating factors remain under investigation. Several lines of evidence implicate the endothelin system (a family of three isopeptides and two specific receptors) as an important modulator of sympathetic activation in the setting of acute coronary syndromes. Such interaction is present in the heart and in the adrenal medulla, whereas less is known on the effects of the endothelin system on the central autonomic network. This article summarizes the current state-of-the-art, placing emphasis on early-phase arrhythmogenesis, and highlights potential areas of future research.

## 1. Introduction

Acute coronary syndromes, the most common clinical manifestation of coronary artery disease, remain a major health-related problem worldwide. Myocardial ischemia in this setting has been linked to the onset of ventricular tachyarrhythmias (VTs) (i.e., ventricular tachycardia and ventricular fibrillation) that often have a lethal outcome. The incidence of sustained VTs complicating acute coronary syndromes is estimated at the range of 10% of cases, although the precise epidemiologic burden is uncertain [1].

The temporal pattern of VTs observed in response to acute ischemic injury has been the subject of preclinical and clinical research. Early and delayed distinct peaks have been described in various species, albeit less discernible in others; despite the limitations in extrapolating these findings to man, clinical data indicate that a bimodal distribution is likely [2]. Regardless of the pathophysiologic background, a distinction between early and delayed VTs is of major clinical significance, as they invariably correspond to pre- and in-hospital phases, respectively.

Prompt revascularization strategies and continuous monitoring in coronary care units have markedly lowered the incidence of delayed VTs and have decreased in-hospital mortality of acute coronary syndromes. By contrast, the prognosis of out-of-hospital, early-phase ischemia-induced VTs remains dismal, despite the continuous refinement of the emergency medical services [2]. Driven by the societal impact of sudden cardiac death, an in-depth understanding of the underlying pathophysiology remains at the core of research efforts, aiming at devising preventive strategies.

## 2. Sympathetic Activation during Myocardial Ischemia

Sympathetic activation in response to acute coronary occlusion has been long recognized as an important contributor to arrhythmogenesis. Following an immediate exocytotic phase, norepinephrine is released from sympathetic nerve terminals in the ischemic myocardium via non-exocytotic mechanisms, with such local release demonstrated in isolated, ex vivo beating preparations [3]. In the in vivo setting, ischemia-induced sympathetic activation entails more complex responses that involve the entire sympatho-adrenal axis. Catecholamines are secreted from chromaffin cells in the adrenal medulla, whereas efferent discharges from the brain stem stimulate myocardial sympathetic nerve endings globally in the heart [4]. Understanding the complex sympathetic activation during acute coronary syndromes is important, based on the spatial variations associated with each arm. More specifically, the effects of circulating and neurally mediated norepinephrine differ, as circulating catecholamines can only reach myocardial sites with adequate perfusion, in contrast to local release that occurs in the synaptic cleft of ischemic areas [5].

Sympathetic activation elicits positive inotropic effects that serve the maintenance of cardiac output but creates a highly arrhythmogenic milieu in the ventricular myocardium. Studies in conscious large-animal models [6] and patients [7] have demonstrated the precedence of VTs by enhanced sympathetic activity, supporting epidemiologic data in patients with acute coronary syndromes complicated by early-phase VTs [8].

Sympathetic activation participates in the genesis of VTs via several mechanisms [1]. It raises the cardiomyocyte resting membrane potential, thereby enhancing automaticity, and induces delayed afterdepolarizations that can lead to triggered activity; ectopic rhythms are sustained by the dispersion of ventricular repolarization that forms areas with inhomogeneous local electrophysiologic properties [9]. The latter mechanism prevails in the setting of myocardial ischemia, due to its opposite actions on the repolarization of the non-ischemic versus the ischemic myocardium, consisting of shortening versus prolongation, respectively [10]. As a result, spatial differences in excitability are enhanced, particularly at the rim of the ischemic zone, thereby creating functional substrates for reentrant circuits.

## 3. The Endothelin System during Acute Coronary Syndromes

The endothelin system consists of a family of three endothelin isopeptides (ET-1, ET-2, and ET-3), produced by numerous cell types, and two specific receptors (ET_A_ and ET_B_) that are widely expressed throughout the body. Endothelin is synthesized and released continuously, but it is also stored in intracellular endothelial storage pools and secreted by exocytosis. Shortly after its discovery, it was shown in animal models [11] and patients [12] that endothelin plasma levels rise shortly after acute coronary occlusion (Figure 1).

The main source of circulating endothelin in the setting of acute coronary syndromes appears to be the ischemic myocardium, as demonstrated in large animal models [13,14], with plasma levels correlating with the incidence of VTs [14]. Prognostic implications were shown clinically [15], indicating that this ubiquitous peptide participates in several pathophysiologic processes during myocardial ischemia. In addition to its vasoconstrictive effects in the coronary circulation, endothelin is implicated in ischemia-induced arrhythmogenesis, both directly and indirectly, the latter mode exerted by modulating sympathetic responses [16].

### 3.1. Direct Effects

Endothelin exerts arrhythmogenic effects in isolated ventricular cardiomyocytes, consisting of enhanced automaticity and early afterdepolarizations [17]. Several cellular mechanisms underlying these actions have been proposed, such as enhanced calcium release from the sarcoplasmic reticulum (acting via inositol trisphosphate receptors), inhibition of delayed rectifier potassium current, or activation of the Na^+^/H^+^ exchanger [18]. Moreover, endothelin may impair the gap junctional coupling of cardiomyocytes, thereby contributing to anisotropic conduction [19]. The latter mechanism has been demonstrated in cellular electrophysiologic studies, but its importance during acute myocardial ischemia is unclear. Of note, preliminary in vivo experiments by our group lend support to these findings, after the analysis of local activation in the ventricular myocardium, by means of multi-electrode array recordings [20]. Based on the potential importance of this mechanism, further investigation is required on the effects of the endothelin system on electrical conduction in the ischemic myocardium.

### 3.2. Indirect Effects

In addition to the direct arrhythmogenic properties of endothelin, several lines of evidence have demonstrated a complex interplay between the endothelin system and sympathetic activation [21]. This is present at the adrenal gland level and at the ventricular myocardial level, with endothelin receptors exerting opposing effects [16].

### 3.3. ET_A_ and ET_B_ Receptors in the Adrenal Gland

The role of endogenous endothelin in catecholamine secretion in response to electrical stimulation was first investigated in isolated, perfused adrenal glands of rats [22]; in these experiments, selective ET_A_-receptor blockade inhibited catecholamine output, whereas pre-treatment with selective ET_B_-receptor blockade abolished this response.

The significance of these findings on arrhythmogenesis during acute coronary syndromes is uncertain. Utilizing the adrenalectomy and ET_B_-deficient rat models, our group examined the contribution of the adrenal medulla on ischemia-induced VTs and the modulatory effects exerted by the endothelin system [23]. Contrasting delayed VTs, we found evidence that circulating catecholamines represent only a minor contribution to early-phase arrhythmogenesis, with functioning ET_B_ receptors in the ventricular myocardium exerting protective effects (Figure 2).

### 3.4. Endothelin Receptors in the Myocardium

The presence of both endothelin receptors in cardiac sympathetic nerve varicosities were first demonstrated in guinea pig hearts and subsequently in other species, including man [24]. In healthy hearts, the endothelin system interferes with exocytotic norepinephrine release, with the ET_A_-mediated inhibition of norepinephrine re-uptake exceeding the ET_B_-receptor-mediated attenuation of norepinephrine release [25]. The endothelin system plays a prominent role also in non-exocytotic norepinephrine release during myocardial ischemia (Figure 3).

Non-exocytotic norepinephrine release in myocardial sympathetic nerve varicosities is enhanced by the activation of ET_A_ receptors via the stimulation of the neuronal Na^+^/H^+^ exchanger. Na^+^ influx in exchange for H^+^ leads to an accumulation of axoplasmic Na^+^, which, in turn, triggers excessive axoplasmic norepinephrine release via the norepinephrine transporter functioning in reverse mode (i.e., transporting norepinephrine from the intracellular to extracellular space). By contrast, ET_B_ receptors decrease norepinephrine overflow, possibly by enhancing nitric oxide production via nitric oxide synthase [21]. Such opposing effects of endothelin receptors on sympathetic nerve endings during myocardial ischemia were elegantly demonstrated in the ex vivo experimental setting [26], with the role of ET_B_ receptors assessed by means of pharmacological blockade and with the use of the subtraction model of ET_B_-deficient rats (Figure 4).

In a series of experiments [27,28,29], our group extrapolated these findings, examining their impact on arrhythmogenesis. Utilizing the in vivo rat model, we reported a lower total duration of VTs during both (early and delayed) phases post-coronary ligation after ET_A_-receptor blockade, confirming its pathophysiologic role [27]. Despite the pathophysiologic value, the clinical significance of these findings is uncertain, due to the scarcity of relevant clinical data. Preliminary observations in patients with posterior-wall myocardial infarction showed neutral effects on delayed VTs after intravenous ET_A_-receptor blockade, but the small sample size precludes firm conclusions [30].

To examine the role of the ET_B_ receptor, we evaluated the effects of dual (ET_A_ and ET_B_) endothelin-receptor blockade in rats in vivo; in these experiments [28], a decreased incidence of VTs was confined to the delayed phase post-ligation, strongly suggesting the beneficial effects of functioning ET_B_ receptors in the ventricular myocardium during the early phase. We further examined the role of ET_B_ receptors on arrhythmogenesis during myocardial ischemia, using wild-type and ET_B_-deficient rats [29]; this protocol circumvents the limitations associated with selectivity issues after pharmacologic blockade of endothelin receptors. In these experiments [29], the critical role of ET_B_ receptors in ameliorating early-phase sympathetic activation and arrhythmogenesis was reiterated; specifically, early-phase arrhythmogenesis in ET_B_-deficient rats was markedly higher than wild-type rats, resulting in excessive mortality during this period (Figure 5).

## 4. Central Autonomic Network

Autonomic responses during acute coronary syndromes include short-term homeostatic feedback mechanisms, followed by the activation of higher centers in the brain [31]. Central actions play an important role in the genesis of ischemia-induced VTs, although autonomic dysfunction varies depending on the location of the ischemic area [32].

### 4.1. Brain Endothelin System

The factors regulating the central autonomic network during myocardial ischemia are complex and remain incompletely understood. In addition to adrenal and myocardial sites, there is growing evidence that the endothelin system also modulates central autonomic inputs, based on its wide distribution in the brain and spinal cord of experimental animals [33] and humans [34]. Indeed, early studies have described potent hemodynamic changes after intracisternal endothelin administration [35]. Although initially attributed to cerebrovascular effects, the non-vascular location pattern of endothelin receptors in the brain points towards neuromodulatory actions [36], which are likely mediated by alterations in calcium influx and neuronal conduction [37].

Anatomical and functional studies have provided further support to the concept of central autonomic control exerted by the brain endothelin system [38]. For example, tyrosine hydroxylase activity measurements have demonstrated the interaction between the endothelin and the olfactory system [39]. Moreover, functional studies using cellular c-fos expression showed the activation of the brainstem after intracerebroventricular endothelin administration, an action mediated by ET_A_ receptors [40]. Lastly, cardiac sympathetic responses were modulated after exogenous endothelin administration in the paraventricular nucleus in rats; these effects were dose-dependent and were prevented by pretreatment with ET_A_-receptor blockade [41].

Although the physiologic role of the endothelin system in the brain is emerging, its effects during myocardial ischemia remain incompletely understood. Our group has provided some insights on this issue in a series of in vivo experiments [42,43,44]. We initially studied wild-type and ET_B_-deficient rats after permanent coronary ligation, with or without pretreatment with the centrally acting sympatholytic agent clonidine; in these experiments, cardiac rhythm was continuously recorded by implantable telemetry devices [42]. We reported important inputs of central sympathetic activation on early-phase arrhythmogenesis, which were attenuated by ET_B_ receptors in the myocardium.

### 4.2. Brain Endothelin System during Myocardial Ischemia

To further assess the role of the brain endothelin system during myocardial ischemia, we examined the effects of intracerebroventricular ET_A_-receptor blockade in rats [43]; this protocol targets the endogenous endothelin system and avoids the confounding effects of exogenous endothelin administration. We reported beneficial effects on delayed arrhythmogenesis post-coronary ligation, whereas infarct size was unchanged. We subsequently extended our observation period to include both early and delayed arrhythmogenic phases post-coronary occlusion [44]. We found decreased sympathetic activity, evidenced by noninvasive indices derived from heart rate variability analysis (Figure 6), with an improved autonomic function associated with a lower incidence of VTs during both phases.

A causative relation between intervention and lower arrhythmogenesis was strengthened by the attenuated regional myocardial repolarization inhomogeneity in the treated group. In the same work [44], we also examined the role of ET_B_ receptors, with or without concurrent ET_A_-receptor blockade. A between-groups comparison indicated that the antiarrhythmic effects were mainly attributed to ET_A_-receptor blockade; however, pharmacologic selectivity issues may be particularly relevant in the brain endothelin system [45], mandating further research on the role of ET_B_ receptors in the brain.

### 4.3. Brain Endothelin System and Vagal Activity

In addition to the sympathetic component, there is now sufficient evidence to suggest that autonomic modulation by the endothelin system also encompasses vagal responses. Early studies have demonstrated the presence of endothelin receptors in the dorsal vagal complex of the brainstem [46], with vagal activation elicited after intracisternal endothelin administration [47]. These findings were subsequently confirmed after selective endothelin microinjections into the dorsal vagal complex of anesthetized rats; this intervention modulated gastric motor, arterial blood pressure, and heart rate, with such effects mediated via ET_A_ receptors [48]. These findings are in keeping with our aforementioned study [44], reporting a moderately enhanced vagal activity after intracerebroventricular ET_A_-receptor blockade.

Taken together, previous studies indicate that the brain endothelin system modulates vagal responses in the setting of myocardial ischemia. The pathophysiologic importance of these observations is substantial, given the salutary vagal effects on cardiomyocyte refractoriness and ventricular fibrillation threshold [49]. However, several electrophysiological aspects of autonomic activation during acute coronary syndromes remain under investigation. For instance, excessive vagal activity can trigger bradycardia-induced VTs, whereas vagal activation without reciprocal sympathetic withdrawal can also be earrhythmogenic [50]. Thus, the effects of the brain endothelin system on vagal activity and arrhythmogenesis during acute coronary syndromes are far from being understood and constitute a subject for future studies.

## 5. Acute Emotional Stress and Arrhythmogenesis during Myocardial Ischemia

Acute emotional stress evokes complex autonomic responses, displaying distinct features from those elicited during physical exertion [51]. A considerable amount of evidence suggests that the endothelin system is also implicated in autonomic responses during emotional stress. In rats, acute psychosocial stress increases plasma endothelin levels [52] and lowers vagal activity [53]. Remarkably, similar findings were reported in response to acute emotional excitement in healthy subjects and in patients with previous myocardial infarction [54]. The origin of circulating endothelin in such cases has been debated, but the vascular endothelium appears to be the most likely cellular source [55].

The implications of these observations on ischemia-induced VTs are poorly defined. Irrespective of the possible causative link between acute emotional stress and coronary artery disease, this clinical setting is common in contemporary society. An example (perhaps in extremis) is given by the high incidence of VTs among patients with implanted defibrillators following the World Trade Center terrorist attack in 2001 [56]. Furthermore, a cohort study [57], examining patients admitted with acute coronary syndrome after intense excitement, reported profoundly increased endothelin (and inflammatory markers, such as monocyte chemoattractant protein-1) when compared to either a reference group or to healthy controls (Figure 7).

Examined collectively, current evidence suggests a pathophysiologic role of the endothelin system in acute coronary syndromes associated with acute emotional stress, but the possible ramifications on early-phase arrhythmogenesis warrant further investigation.

## 6. Conclusions

Early-phase VTs in the setting of acute coronary syndromes often lead to sudden cardiac death and remain an important health-related problem worldwide. Myocardial ischemia induces major changes that favor the onset of VTs via all known arrhythmogenic mechanisms. This milieu is further altered by sympathetic activation, consisting of local and systemic catecholamine release, each with a distinct electrophysiologic impact. The endothelin system augments both processes via ET_A_ receptors, whereas ET_B_ receptors in the myocardium ameliorate local early-phase sympathetic activation and arrhythmogenesis.

Acute coronary syndromes are often accompanied by autonomic dysfunction that contributes to VTs, but the underlying pathophysiology is complex and incompletely understood. The role of the brain endothelin system in modulating sympathetic and vagal activity has recently emerged, particularly in cases of acute emotional stress preceding myocardial ischemia. The interaction between the endothelin system and the autonomic nervous system is currently under investigation, with a view towards implementing therapeutic strategies that will lower the incidence of sudden cardiac death.

## Figures and Tables

**Figure 1 life-12-01627-f001:**
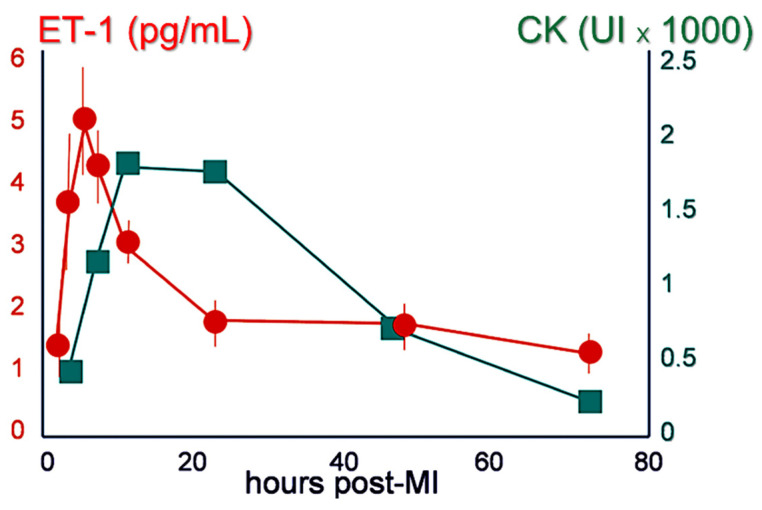
Endothelin-1 (ET-1) plasma levels (**red**) increase sharply shortly after acute coronary occlusion; creatine kinase (CK) levels shown (**green**) for comparison. Reprinted with permission from ref. [12]. Copyright 1991, American College of Cardiology Foundation, Published by Elsevier Inc.

**Figure 2 life-12-01627-f002:**
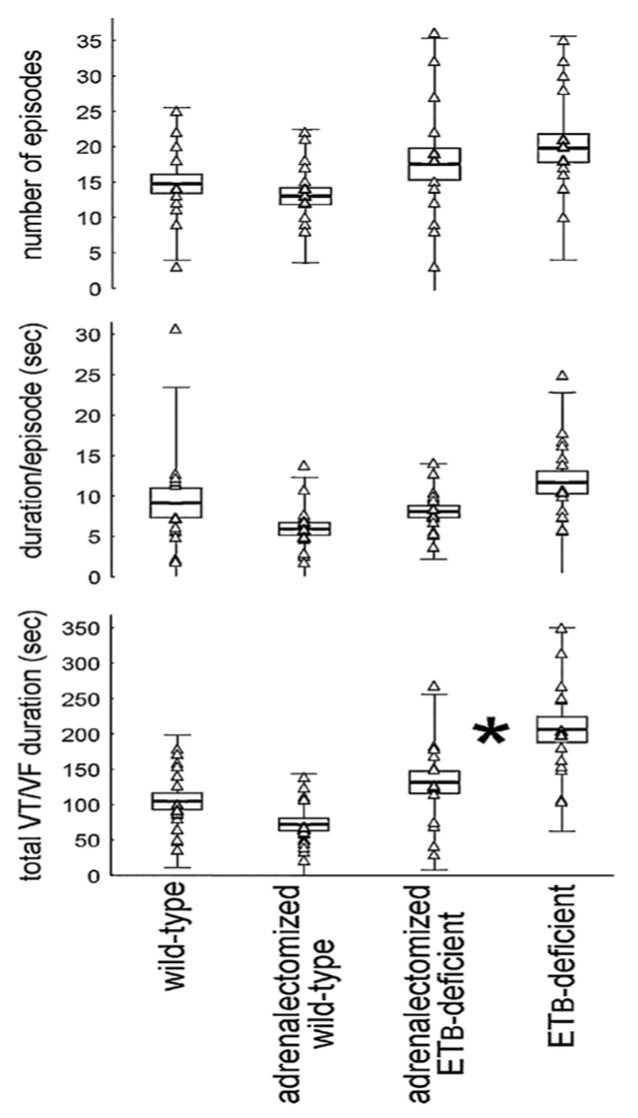
Early-phase arrhythmogenesis, i.e., within the first hour after coronary ligation in rats. Adrenalectomy did not affect ventricular tachyarrhythmias in wild-type rats, indicating minor contribution of circulating catecholamines. However, the total duration of ventricular tachycardia (VT) and fibrillation (VF) episodes was shorter in adrenalectomized ET_B_-deficient rats, suggesting protective effects of functioning ET_B_ receptors in the ventricular myocardium (* denotes significant difference). Reprinted from ref. [23].

**Figure 3 life-12-01627-f003:**
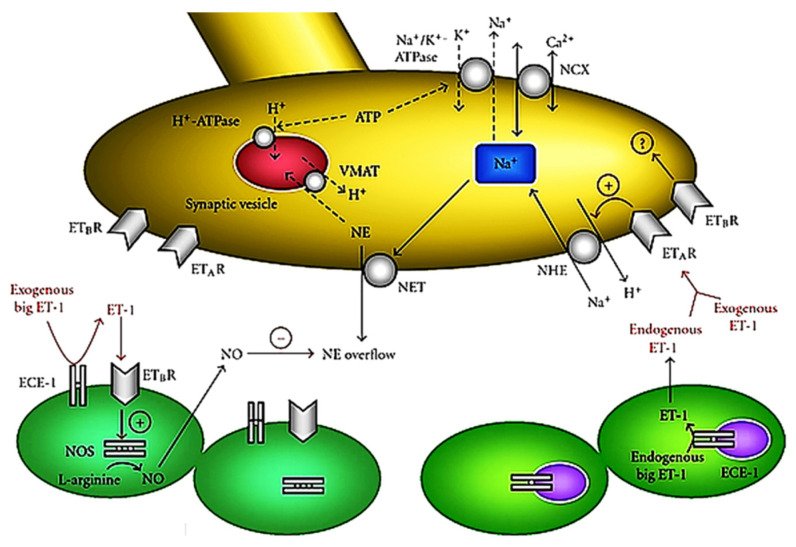
Interaction between the endothelin (ET) system and norepinephrine (NE) release in the ischemic myocardium. Reprinted from ref. [21].

**Figure 4 life-12-01627-f004:**
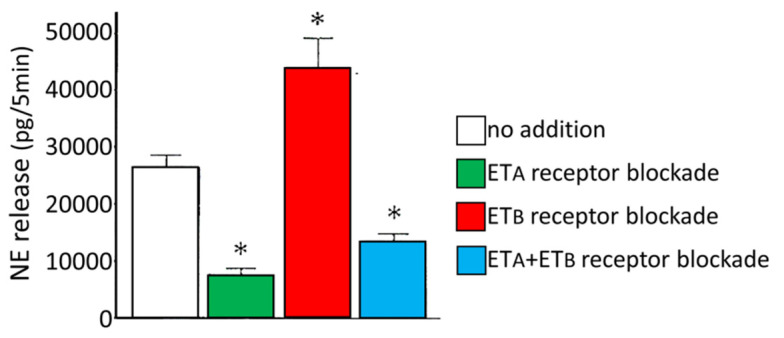
Opposing effects of ET_A_- and ET_B_-receptor blockade on norepinephrine (NE) overflow in ischemic rat hearts (* denote significant differences compared to the *no addition* group). Reprinted with permission from ref [26]. Copyright 2005, Wolters Kluwer Health.

**Figure 5 life-12-01627-f005:**
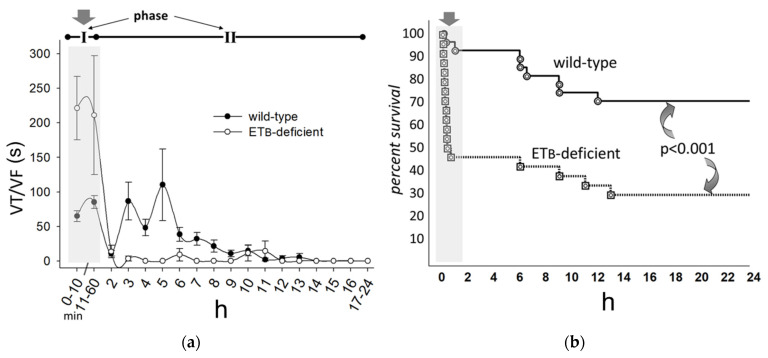
(**a**) ET_B_-deficient rats displayed higher incidence of early-phase (phase I) ventricular tachyarrhythmias (VTs) (arrow); (**b**) As a result, there was excessive mortality during this period (arrow), strongly suggesting a prominent role of functioning ET_B_ receptors. Reprinted with permission from ref. [29]. Copyright 2009, Springer-Verlag.

**Figure 6 life-12-01627-f006:**
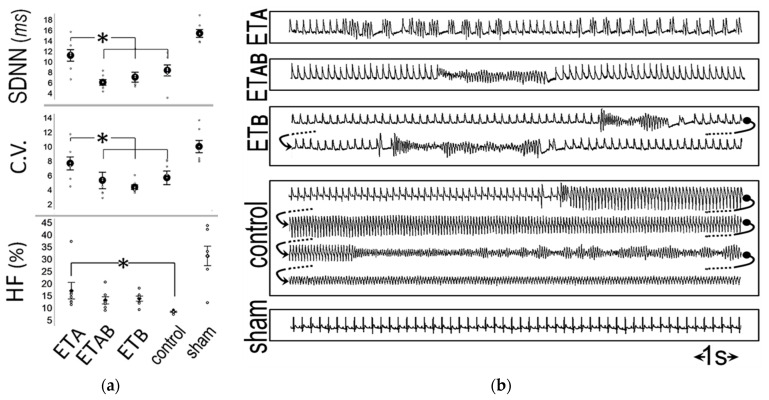
(**a**) In rats, intracerebroventricular ET_A_-receptor blockade improved autonomic (sympathetic and vagal) indices, i.e., the standard deviation of successive RR-intervals (SDNN), the coefficient of variance (C.V.), and the high frequency (HF) component of heart rate variability (* denote significant differences) (**b**) Examples of telemetry recordings displaying lower arrhythmogenesis after intracerebroventricular ET_A_-receptor blockade. Reprinted with permission from ref. [44]. Copyright 2019 Elsevier Inc.

**Figure 7 life-12-01627-f007:**
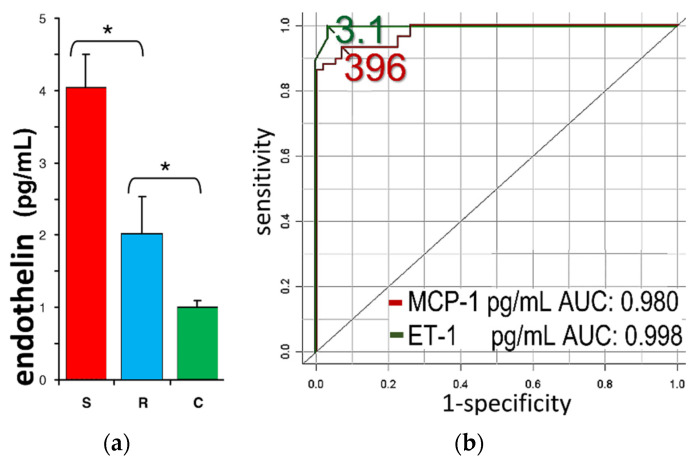
(**a**) Serum levels of endothelin-1 were higher in patients admitted with acute coronary syndromes after acute emotional stress (S, red bar) than reference patients with acute coronary syndrome without preceding emotional stress (R, blue bar) or healthy controls (C, green bar); * denote significant differences. (**b**) Receiver-operating characteristic curves (with areas under the curve, AUC) for the assessment of endothelin-1 (ET-1) (green line) and monocyte chemoattractant protein (MCP)-1 (red line) as specific markers of stress-associated acute coronary syndrome. Reprinted with permission from ref. [57]. Copyright 2010 American College of Cardiology Foundation. Published by Elsevier Inc.

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
