# Peer review of "Endothelin System and Ischemia-Induced Ventricular Tachyarrhythmias"

_life, 2022, doi:10.3390/life12101627_

Round 1

Reviewer 1 Report

The manuscript discussed about acute coronary syndrome that are often accompanied by autonomic dysfunction that contribute to VTs. The  pathophysiology and is complex and incompletely understood and this manuscript try to address the concern.

Author Response

Thank you for your favourable comments regarding the submitted manuscript. Minor corrections to the usage of the English language have been made.

Reviewer 2 Report

very nice, very precise, and detailed review about the role endothelin system on ventricular arrhythmia after acute coronary syndrome.

the review has a potential benefit for the readers since it may enlighten the mechanisms related to ventricular arrhythmia and endothelin system interaction.

Author Response

Thank you for your favourable comments regarding the submitted manuscript. Corrections regarding the use of the English language have been made to the revised version.

Reviewer 3 Report

The manuscript "Endothelin system and ischemia-induced ventricular tachyarrhythmias" by Eleni-Taxiarchia Mouchtouri, Thomas Konstantinou, Panagiotis Lekkas, and Theofilos Kolettis presents a very thorough and highly interesting review on the factors and mechanisms involved in the genesis of ventricular tachyarrhythmias during coronary ischemia (i.e. myocardial infarction). Insights from the past decades that deciphered the endothelin system as a central player are shared, without leaving the realm of what is known and clearly stating where processes are not yet fully understood. As Kolettis and his team have been working on this closely in the past years, many publications are self-citations - none of which are deemed inappropriate. Every reference is carefully selected and meaningful. Overall, this was a very worthwhile read: Even though the matter is complex, the authors succeed in succinctly presenting these data and understandings derived therefrom.

The only remark that I have is: It seems as if the affiliations were not complete. Please amend these and provide information on the corresponding author.

Bottom line: I very much enjoyed readaing this manuscript which I esteem. It is of high-quality and a valuable addition to the journal. I wholeheartedly recommend publication.

Author Response

Thank you for your favourable comments regarding the submitted manuscript. The affiliations of the authors have been amended in the revised manuscript and information on the corresponding author has been added.